# Semantic or Covariate? A Study on the Intractable Case of Out-of-Distribution Detection

## Abstract

The primary goal of out-of-distribution (OOD) detection tasks is to identify inputs with semantic shifts, i.e., if samples from novel classes are absent in the in-distribution (ID) dataset used for training, we should reject these OOD samples rather than misclassifying them into existing ID classes. However, we find the current definition of "semantic shift" is ambiguous, which renders certain OOD testing protocols intractable for the post-hoc OOD detection methods based on a classifier trained on the ID dataset. In this paper, we offer a more precise definition of the Semantic Space and the Covariate Space for the ID distribution, allowing us to theoretically analyze which types of OOD distributions make the detection task intractable. To avoid the flaw in the existing OOD settings, we further define the "Tractable OOD" setting which ensures the distinguishability of OOD and ID distributions for the post-hoc OOD detection methods. Finally, we conduct several experiments to demonstrate the necessity of our definitions and validate the correctness of our theorems.

## 1 Introduction

Out-of-distribution (OOD) detection has gained increasing attention in recent years due to its crucial role in ensuring algorithmic reliability. While deep learning models excel on in-distribution (ID) samples, i.e., those belonging to the same classes as the training set, these models often struggle when faced with unseen OOD classes Hendrycks & Gimpel (2017). For example, a model trained on a fruit classification dataset may generate arbitrary and incorrect results when presented with an image of an animal. Their primary objective of OOD detection is to enable models to identify inputs that fall outside the ID distribution and reject them, rather than blindly classifying them into existing ID classes, thus mitigating potential security risks. Existing OOD detection research has made significant progress Hendrycks & Gimpel (2017); Liang et al. (2018); Lee et al. (2018); Hendrycks et al. (2019); Hsu et al. (2020); Liu et al. (2020); Huang et al. (2021). Particularly in the more extensively studied and widely adopted post-hoc OOD detection methods Hendrycks & Gimpel (2017); Liang et al. (2018); Lee et al. (2018); Liu et al. (2020); Huang et al. (2021); Sun et al. (2022); Sun & Li (2022); Song et al. (2022); Djurisic et al. (2023); Kim et al. (2023); Xu et al. (2024), they leave the original model training unchanged and instead utilize the features or prediction outputs of the test data to assess whether a sample is OOD, making the OOD detection process highly efficient.

However, we find the current definition of OOD detection setting to be flawed, which renders certain OOD testing protocols intractable for the post-hoc OOD detection methods and thus undermines the significance of the OOD detection task. In existing OOD literature, OOD samples are typically defined as data exhibiting semantic shifts relative to the training set Yang et al. (2024a), in which the concept of "semantic shift" lacks clear definition and boundaries. For the example shown in Fig. 1, if an ID dataset contains classes of dog breeds like "collie", "husky", and "chihuahua", while the OOD data includes a novel breed "beagle", the OOD detection task is well-defined as the model can identify the OOD class by the "breed-specific feature" it learns from the "breed-separated" training setup. However, if the model is trained on an object classification dataset that aggregates the different dog breeds into a single class "dog", while the OOD data still includes the novel breed "beagle" that is absent in the ID data, whether to consider this variation on dog breeds a "semantic

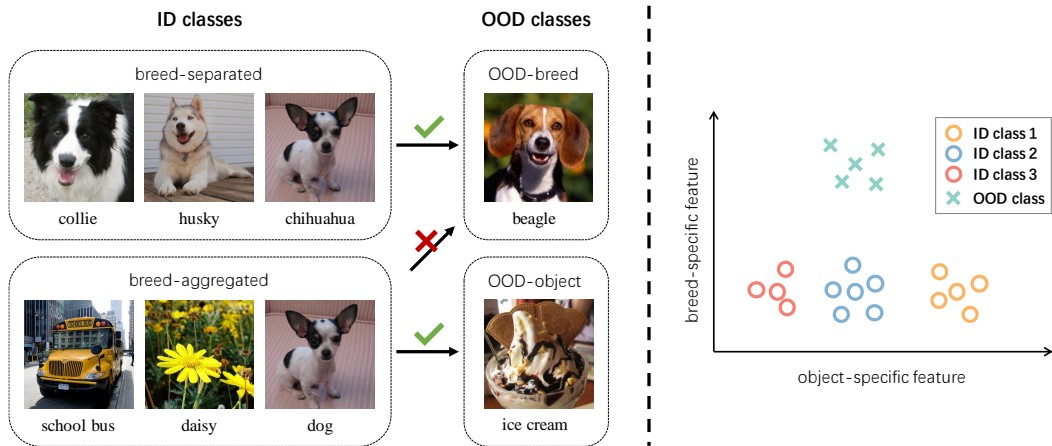

(a) Two Training Setups vs. Two Testing Protocols         (b) Feature Space of the Intractable Case

Figure 1: An illustration of the intractable OOD detection setting. (a) shows two training setups and two testing protocols. After being trained under the "breed-aggregated" setup, the model will struggle to identify a novel dog breed in the "OOD-breed" testing protocol. (b) demonstrates the feature distribution in this intractable case, where the ID classes are well-separated along the object-specific feature, while the OOD class differs from the ID classes only along the breed-specific feature dimension.

shift" becomes ambiguous under the current OOD definition. We find that distinguishing between ID and OOD dog breeds in this case will become intractable because the classifier, when trained under the "breed-aggregated" setup, primarily focuses on "object-specific feature" and will see the "breed-specific feature" as covariate factors, as can be seen in Fig. 1(b). This illustrates that, despite the testing OOD data being from a novel class different from the ID data as defined in existing OOD literature, it remains unsolvable for post-hoc OOD detection methods to identify the OOD data. Therefore, we believe that the scope of "semantic shift" should be more precisely defined based on the characteristics of the training data, rather than by the object classes as in existing works.

To address this issue, we first provide a more precise definition for the Semantic Space and the corresponding Covariate Space in this paper. Specifically, we construct the Semantic Space using a linear span of representative feature vectors from each ID class. These representative feature vectors are the centers of each ID class's distribution, capturing the overall characteristics of the samples within that class. With the defined Semantic Space, we can represent the Covariate Space using direct sum decomposition and then identify the range of shifts that a model cannot distinguish when it is exposed solely to the ID data. Our theoretical analysis shows that: "if two classes do not exhibit any shift in the Semantic Space, they will be indistinguishable by a post-hoc OOD detection model based on a classifier trained only on the ID dataset". This theorem well explains the difficulty of distinguishing ambiguous OOD classes, such as the intractable case in Fig. 1, where the shift occurs only within the Covariate Space for the model trained under the "breed-aggregated" setup. Furthermore, we propose the "Tractable OOD" setting, where the OOD shift must occur within the Semantic Space we define, thereby mitigating the issue of undetectability. Finally, extensive experiments are conducted to effectively demonstrate the necessity of our definitions and the validity of the theoretical analysis.

Our main contributions can be summarized as follows:

- We provide more accurate definitions for the Semantic Space and the Covariate Space within the OOD detection tasks. Based on these definitions, we introduce the "Tractable OOD" setting, which ensures the OOD detection task is tractable.

- We present a theoretical analysis that proves the intractability of detecting OOD classes that exhibit no shift within the defined Semantic Space. The analysis enhances our understanding of the flaw in the current OOD detection definition.

- We conduct several experiments, further validating the necessity of our definitions and the correctness of our theorems.

## 2 PROBLEM SETUP

Below, we provide the notions for Out-of-Distribution (OOD) detection as described in most existing works Morteza & Li (2022); Du et al. (2024b); Yang et al. (2024b).

**Labeled ID distribution.** Let the input space $\mathcal{X}$ be a d-dimensional space $\mathbb{R}^d$. The label space for ID data is denoted as $\mathcal{Y}_I = \{y_1, ..., y_k\}$. The ID data is independently and identically sampled from the joint distribution $(\boldsymbol{x}, y) \sim \mathcal{P}_I$ defined over $\mathcal{X} \times \mathcal{Y}_I$.

**OOD distribution.** The input of the OOD distribution is also from the space $\mathcal{X}$ but with OOD label $\mathcal{Y}_O = \{y_o\}$, where $y_o \notin \mathcal{Y}_I$. The OOD data is drawn from the joint distribution $(\boldsymbol{x}, y) \sim \mathcal{P}_O$ defined over $\mathcal{X} \times \mathcal{Y}_O$. For OOD detection, only the input $\boldsymbol{x}$ is available for use by the models.

However, the definition above lacks explicit connections between the input and the label, complicating the analysis of the model's training process on these data. In real-world tasks, samples sharing the same label inevitably exhibit common features, e.g., school buses are typically yellow. Therefore, following the previous setup in Morteza & Li (2022), we assume that the input of each class can be modeled as a Gaussian distribution in this paper. To further simplify the theoretical analysis, we also assume that all the classes in the input space are linearly separable, allowing the model trained on these data to be treated as a linear classifier. The modified definition of the OOD detection task is as follows:

**OOD Detection in Gaussians.** For the class-conditional ID distribution $(\boldsymbol{x}, y_i) \sim \mathcal{P}_I|y_i$, we assume the input follows a Gaussian distribution $\boldsymbol{x} \sim \mathcal{N}(\boldsymbol{\mu}_i, \boldsymbol{I})$, where $\boldsymbol{\mu}_i$ denotes the representative feature vector. In addition, all the representative feature vectors in the input space are linearly separable. Similarly, the OOD input follows a Gaussian distribution $\mathcal{N}(\boldsymbol{\mu}_o, \boldsymbol{I})$ with $\boldsymbol{\mu}_o$ as the representative feature vector. There exists a shift between OOD and ID distributions, with the minimum distance between them denoted as $\delta$:

$$\|\boldsymbol{\mu}_o - \boldsymbol{\mu}_i\|_2^2 \geq \delta, i \in \{1, ..., k\}. \tag{1}$$

*Remark.* Defining the OOD problem as described above simplifies the analysis by avoiding the complexities introduced by nonlinear transformations during feature extraction in deep neural network training. Our experimental results demonstrate that this simplification does not compromise the validity of the conclusions presented in the subsequent sections.

Building on the OOD detection setup above, we now introduce our definitions of the Semantic Space and the Covariate Space for the ID distribution.

**Definition 1** (Semantic Space). The Semantic Space $\mathcal{S}$ based on ID distributions is

$$\mathcal{S} = span(\{\boldsymbol{\mu}_1 - \boldsymbol{\mu}_2, ..., \boldsymbol{\mu}_k - \boldsymbol{\mu}_{k-1}\}). \tag{2}$$

*Remark.* The definition of the Semantic Space relies on the differences between the representative feature vectors. This approach ensures that each class's representative feature vector has distinct components in the Semantic Space while removing any common elements shared across all representative feature vectors.

**Definition 2** (Covariate Space). The Covariate Space $\mathcal{C}$ is defined using direct sum decomposition

$$\mathcal{X} = \mathcal{S} \oplus \mathcal{C}. \tag{3}$$

*Remark.* It is important to note that the defined Covariate Space is determined by the Semantic Space, meaning the Covariate Space also depends on the ID data in our definition. For instance, in object classification tasks, the image background is considered part of the Covariate Space, whereas it is not in scene recognition tasks.

To better understand the defined Semantic Space and Covariate Space, we present the following proposition which explains the practical significance of the definition.

**Proposition 1.** *After decomposing the representative feature vector of any ID distribution into components within the Semantic Space and the Covariate Space, the feature component within the Covariate Space remains a constant vector $c_{const}$*

$$\boldsymbol{\mu}_i = \boldsymbol{s}_i + \boldsymbol{c}_i, \boldsymbol{s}_i \in \mathcal{S}, \boldsymbol{c}_i \equiv \boldsymbol{c}_{const} \in \mathcal{C}, i \in \{1, ..., k\}. \tag{4}$$

See proof in Appendix A.1. The proposition shows our definition ensures that the covariate components of the representative feature vectors remain constant across different ID classes. In other words, variations in a sample's feature vector within the Covariate Space do not influence the determination of its class in the ID label space.

With the definitions of Semantic Space and Covariate Space established, we can briefly analyze the intractable case in Fig. 1. Under the "breed-separated" setup, where different classes represent distinct dog breeds, the differences between the representative feature vectors of each class capture the distinctions between breeds. According to our definition, the "breed-specific feature" belongs to the Semantic Space. However, under the "breed-aggregated" setup, since different classes represent different objects, the differences between the representative feature vectors do not reflect breed distinctions, placing the "breed-specific feature" within the Covariate Space. Thus, we conclude that the range of the Semantic Space and the Covariate Space depends on the training setups, and post-hoc OOD detection models should be sensitive only to shifts within the Semantic Space. In the next section, we provide a concrete theoretical analysis to support this conclusion.

## 3 THEORETICAL ANALYSIS

In this section, we investigate the classifier trained on the ID distribution to enable a theoretical analysis of the OOD detection process in post-hoc algorithms based on this classifier.

As discussed in the previous section, we can use a linear classifier $f : \mathcal{X} \to \mathcal{Y}_I$ with a weight matrix $\boldsymbol{W}$ to represent the model used in the post-hoc algorithms. The prediction probabilities $p_i$ for each class are then obtained through a softmax layer

$$f(\boldsymbol{x}) = softmax(\boldsymbol{W}\boldsymbol{x}) = [p_1(\boldsymbol{x}), ..., p_k(\boldsymbol{x})]^\top,$$
$$p_i(\boldsymbol{x}) = \frac{\exp\left(\boldsymbol{W}_{i,:}\boldsymbol{x}\right)}{\sum_{j=1}^k \exp\left(\boldsymbol{W}_{j,:}\boldsymbol{x}\right)}, i \in \{1, ..., k\}. \tag{5}$$

Before conducting the theoretical analysis, we introduce two mild assumptions to simplify the analysis process.

**Assumption 1.** *Throughout the entire training process, the expected prediction probability of the linear classifier for each class is equal*

$$\mathbb{E}_{\boldsymbol{x}}[p_i(\boldsymbol{x})] = \frac{1}{k}, i \in \{1, ..., k\}. \tag{6}$$

Assumption 1 requires a balanced ID distribution across all classes, ensuring that the classification model does not develop any inherent bias toward a particular class during training.

**Assumption 2.** *The sign of the covariance between the prediction probability for a specific class $i$ and the $j^{th}$ element of the input vector $\boldsymbol{x}$ matches that of its corresponding weight $\boldsymbol{W}_{ij}$*

$$\boldsymbol{W}_{ij} \cdot Cov(p_i(\boldsymbol{x}), \boldsymbol{x}_j) \geq 0, i \in \{1, ..., k\}, j \in \{1, ..., d\}. \tag{7}$$

Assumption 2 is intuitively straightforward: when $\boldsymbol{W}_{ij}$ is positive, $p_i$ defined in Eq. (5) is a monotonically increasing function of $\boldsymbol{x}_j$, resulting in a covariance that is likely positive. Conversely, when $\boldsymbol{W}_{ij}$ is negative, the covariance is likely negative. We propose Assumption 2 because calculating the analytical solution for the covariance is highly complex due to the involvement of the softmax function in $p_i$. Therefore, Assumption 2 is introduced for simplification. We validate the above two assumptions in our experiments, with the results presented in Appendix C.

After presenting the two assumptions above, we introduce the following important proposition, which provides a formal conclusion regarding the ability of a linear classifier trained on the ID

distribution to perceive features in the defined Semantic Space and Covariate Space. The proof is in Appendix A.3.

**Proposition 2.** *Assume Assumption 1 and Assumption 2 hold, the weight matrix $\boldsymbol{W}$ of the linear classifier $f$, after being trained on the ID distribution, can always be decomposed into a simplified weight matrix $\tilde{\boldsymbol{W}}$ and an orthogonal matrix $\boldsymbol{Q}$. The last $d - r$ columns of $\tilde{\boldsymbol{W}}$ are all zero vectors, and the first $r$ rows of $\boldsymbol{Q}$ form an orthogonal basis for the Semantic Space $S$. $r$ is the rank of the Semantic Space $S$.*

$$\boldsymbol{W} = \tilde{\boldsymbol{W}}\boldsymbol{Q},$$
$$s.t. \quad \tilde{\boldsymbol{W}}_{:,r+1} = ... = \tilde{\boldsymbol{W}}_{:,d} = \boldsymbol{0}, \tag{8}$$
$$span(\{\boldsymbol{Q}_{1,:}, ..., \boldsymbol{Q}_{r,:}\}) = \mathcal{S}.$$

*Remark.* The proposition shows that part of the weight matrix $\boldsymbol{W}$ becomes zero after a specific orthogonal transformation. This indicates that a particular subspace of the input vector has no effect on the classifier's output. Specifically, according to the proof in Appendix A.3, the Covariate Space component of the input corresponds to the zero weights in the simplified weight matrix $\tilde{\boldsymbol{W}}$, implying that the classifier trained on the ID distribution is insensitive to the Covariate Space component.

After deriving the above proposition, we can further present the following theorem, which proves the indistinguishability between certain input distributions. The proof can be found in Appendix A.4.

**Theorem 1.** *For any two classes of data with Gaussian distributions $\mathcal{N}(\boldsymbol{\mu}_a, \boldsymbol{I})$ and $\mathcal{N}(\boldsymbol{\mu}_b, \boldsymbol{I})$ within the input space $\mathcal{X}$, if their representative feature vectors $\boldsymbol{\mu}_a$ and $\boldsymbol{\mu}_b$ are identical in the Semantic Space $\mathcal{S}$, they are indistinguishable to the linear classifier $f$ trained on the ID distribution*

$$if \quad proj_{\mathcal{S}}(\boldsymbol{\mu}_a) = proj_{\mathcal{S}}(\boldsymbol{\mu}_b),$$
$$KL(f(\mathcal{N}(\boldsymbol{\mu}_a, \boldsymbol{I}))||f(\mathcal{N}(\boldsymbol{\mu}_b, \boldsymbol{I}))) = 0. \tag{9}$$

*Remark.* The theorem shows that the model's output distribution is only related to the Semantic Space components of the representative feature vector of the input distribution and is independent of the corresponding Covariate Space components.

Finally, we can derive the following corollary, which proves the intractability of the post-hoc OOD detection models in detecting certain samples, like the case in Fig. 1.

**Corollary 1.** *If the representative feature vector $\boldsymbol{\mu}_o$ of an OOD distribution $\mathcal{N}(\boldsymbol{\mu}_o, \boldsymbol{I})$ is the same as any of the ID distribution in the Semantic Space $\mathcal{S}$, it becomes intractable for any post-hoc OOD detection method to identify the class.*

*Remark.* The corollary can be easily derived from Theorem 1. Since the representative feature vector of the OOD distribution is the same as that of a certain ID distribution in the Semantic Space, Theorem 1 implies that the output distributions of these two after passing through the linear classifier $f$ trained on the ID distribution will be identical. Consequently, any post-hoc OOD detection method based on the classifier's output will be unable to distinguish between these two distributions, rendering the OOD detection in this case an intractable problem.

Given the intractability under the current ambiguous definition of OOD classes, as demonstrated in Corollary 1, we propose a more rigorous definition of OOD setting as below, ensuring the OOD detection task is tractable for the post-hoc model trained on the ID dataset.

**Definition 4** (Tractable OOD). We define the input from a Gaussian $\mathcal{N}(\boldsymbol{\mu}_o, \boldsymbol{I})$ a $\delta$-OOD distribution if its representative feature vector $\boldsymbol{\mu}_o$ is greater than $\delta$ away from the vector of any ID distributions in the Semantic Space $\mathcal{S}$

$$\|\boldsymbol{s}_o - \boldsymbol{s}_i\|_2^2 \geq \delta, i \in \{1, ..., k\},$$
$$where \quad \boldsymbol{s}_o = proj_{\mathcal{S}}(\boldsymbol{\mu}_o), \boldsymbol{s}_i = proj_{\mathcal{S}}(\boldsymbol{\mu}_i). \tag{10}$$

The definition ensures that the representative feature vector of the OOD distribution differs from any ID distribution within the Semantic Space, thereby avoiding the situation in Corollary 1, where the OOD output distribution is identical to one of the ID output distribution. Additionally, we quantify the degree of the shift using $\delta$. Generally, a larger $\delta$ increases the distance between the OOD and the known ID distributions, making it easier for models to identify the OOD samples.

Table 1: AUROC (%) of three OOD detection methods in the Synthetic Data experiments

| | w/o shift in Semantic Space | | | with shift in Semantic Space | | |
|---|---|---|---|---|---|---|
| $s_o$ | $[\sigma, \sigma, 0, 0]^\top$ | | | $[0, \sigma, 0, 0]^\top$ | | |
| $c_o$ | $[0, 0, \sigma, \sigma]^\top$ | $[0, 0, -\sigma, \sigma]^\top$ | $[0, 0, -\sigma, -\sigma]^\top$ | $[0, 0, \sigma, \sigma]^\top$ | $[0, 0, -\sigma, \sigma]^\top$ | $[0, 0, -\sigma, -\sigma]^\top$ |
| MSP Hendrycks & Gimpel (2017) | 50.8 | 50.7 | 51.4 | 79.3 | 79.7 | 79.6 |
| EBO Liu et al. (2020) | 51.0 | 51.1 | 51.4 | 73.4 | 74.8 | 74.8 |
| GradNorm Huang et al. (2021) | 51.5 | 50.6 | 51.8 | 75.3 | 75.0 | 76.4 |

## 4 EXPERIMENTS

We conduct two experiments to demonstrate the necessity of our definitions and the correctness of the proposed theorem. The first experiment is based on data synthesized from Gaussian distributions in a low-dimensional feature space, allowing us to better understand the optimization process of the linear classifier and the proposed Proposition 2. The second experiment considers a more practical scenario, using data from ImageNet-1K Deng et al. (2009) as the training set and a ResNet-18 He et al. (2016) as the classifier. This experiment aims to demonstrate that our simplification regarding "OOD Detection in Gaussians" does not compromise the validity of our theoretical analysis.

We use AUROC as our metric, which is commonly employed in existing OOD detection studies Yang et al. (2021; 2022; 2023; 2024b), to evaluate the performance of OOD detection methods in our experiments. For specific training hyperparameter settings across all experiments, please refer to Appendix B.

### 4.1 SYNTHETIC DATA

**Experimental settings.** We adopt a four-dimensional vector space as the input space $\mathcal{X}$ in the "Synthetic Data" experiments. For easier understanding, the first two dimensions are referred to as the Semantic Space, while the remaining two are designated as the Covariate Space during data generation, keeping the two spaces unscrambled. Additionally, we also conduct experiments in the scrambled case, with the results presented in Appendix C.

We set four ID classes for training and identify the representative feature vectors as follows:

$$\boldsymbol{\mu}_1 = [\sigma, \sigma, \sigma, \sigma]^\top, \quad \boldsymbol{\mu}_2 = [\sigma, -\sigma, \sigma, \sigma]^\top,$$
$$\boldsymbol{\mu}_3 = [-\sigma, \sigma, \sigma, \sigma]^\top, \quad \boldsymbol{\mu}_4 = [-\sigma, -\sigma, \sigma, \sigma]^\top, \tag{11}$$

where $\sigma$ is a constant that determines the position of the representative feature vectors.

During training, data is randomly sampled from the ID distribution $\mathcal{N}(\boldsymbol{\mu}_i, \boldsymbol{I})$ with label $i \in \{1, 2, 3, 4\}$. The training objective is to optimize a linear classifier $f$ with a weight matrix $\boldsymbol{W} \in \mathbb{R}^{k \times d}$, where $k = 4$ denotes the number of classes and $d = 4$ represents the input dimension.

During testing, data is randomly sampled from the OOD distribution $\mathcal{N}(\boldsymbol{\mu}_o, \boldsymbol{I})$ as well as from one of the ID distribution $\mathcal{N}(\boldsymbol{\mu}_1, \boldsymbol{I})$. We investigate the capacity of the OOD detection model to distinguish between these two distributions. Specifically, the components of the OOD representative feature vectors $\boldsymbol{\mu}_o$ are chosen as follows:

$$\boldsymbol{\mu}_o = \boldsymbol{s}_o + \boldsymbol{c}_o,$$
$$\boldsymbol{s}_o \in \{[\sigma, \sigma, 0, 0]^\top, [0, \sigma, 0, 0]^\top\}, \tag{12}$$
$$\boldsymbol{c}_o \in \{[0, 0, \sigma, \sigma]^\top, [0, 0, -\sigma, \sigma]^\top, [0, 0, -\sigma, -\sigma]^\top\}.$$

In the Semantic Space, no shift occurs when $\boldsymbol{s}_o = [\sigma, \sigma, 0, 0]^\top$, whereas a shift is introduced when $\boldsymbol{s}_o = [0, \sigma, 0, 0]^\top$. In the Covariate Space, there is no shift when $\boldsymbol{c}_o = [0, 0, \sigma, \sigma]^\top$, but shifts occur when $\boldsymbol{c}_o$ takes on the values of the other two vectors.

**Performance of OOD detection methods.** Since the input vectors are directly provided as ground truth features for training the linear classifier, we exclude OOD detection methods that might involve manipulation of the feature space to avoid potential "cheating". The results of three tested methods are shown in Table 1. When the OOD representative feature vector does not exhibit shifts

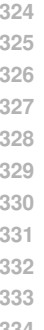
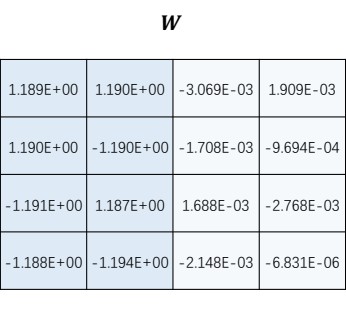
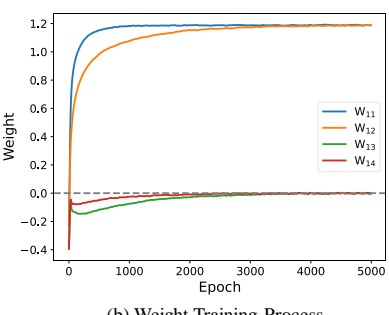

(a) Final Weight Matrix        (b) Weight Training Process

Figure 2: Visualization of the weight matrix $W$ in the linear classifier: (a) the final optimized weight matrix, (b) the variations of the weights for each input dimension corresponding to the first class.

Table 2: AUROC (%) of OOD detection methods across different shift degree $\delta$

| $\delta$ | $0.25\sigma$ | $0.50\sigma$ | $0.75\sigma$ | $1.00\sigma$ |
|---|---|---|---|---|
| $s_o$ | $[0.75\sigma, \sigma, 0, 0]^\top$ | $[0.5\sigma, \sigma, 0, 0]^\top$ | $[0.25\sigma, \sigma, 0, 0]^\top$ | $[0, \sigma, 0, 0]^\top$ |
| $c_o$ | $[0, 0, \sigma, \sigma]^\top$ | | | |
| MSP Hendrycks & Gimpel (2017) | 59.1 | 71.7 | 77.0 | 81.0 |
| EBO Liu et al. (2020) | 59.4 | 67.9 | 72.4 | 76.0 |
| GradNorm Huang et al. (2021) | 58.0 | 67.2 | 73.4 | 76.8 |

in the Semantic Space (the three left columns of the results), all the OOD detection methods fail to distinguish between the OOD and ID distributions, with detection AUROC around 50%, regardless of whether there is a shift in the Covariate Space. This indicates that the OOD detection task becomes intractable for these post-hoc methods, which aligns with our proposed Corollary 1. In contrast, when the OOD representative feature vector includes a shift in the Semantic Space (the three right columns of the results), the OOD detection methods can effectively identify the OOD samples, achieving high AUROC scores.

Additionally, both sets of experimental results clearly show that the changes in the Covariate Space of the OOD representative feature vector do not influence the performance of the OOD detection methods. The findings strongly support our Theorem 1, which asserts that when the representative feature vectors of two classes share identical components in the Semantic Space, the post-hoc OOD detection model will yield indistinguishable outputs for them.

**Weight matrix of the classifier.** The final optimized weight matrix $W$ in the linear classifier is shown in Fig. 2(a). The last two columns of the optimized weight matrix are nearly zero vectors, which confirms the validity of Proposition 2.

To further investigate the optimization process of the weight matrix during training, we display the variations of the weights for each input dimension corresponding to the first ID class, as shown in Fig. 2(b). The figure shows that the absolute values of the weights corresponding to the Semantic Space ($W_{11}, W_{12}$) gradually increase, while those associated with the Covariate Space ($W_{13}, W_{14}$) gradually converge to zero. This result aligns with the conclusion reached in the proof of Proposition 2 in Appendix A.3.

**Effect of the shift degree $\delta$.** We conduct an experiment to analyze the impact of the distance $\delta$ in our definition of "Tractable OOD" on OOD detection performance. The results, presented in Table 2, show that as $\delta$ increases, the detection AUROC of the OOD detection models improves. Conversely, when $\delta$ is too small, the OOD detection models struggle to differentiate OOD samples, indicating that the OOD detection test is not well-defined in such cases. The findings highlight the crucial role of the $\delta$ parameter in determining the tractability of an OOD detection test setting.

## 4.2 IMAGENET DOGS

**Experimental Settings.** Since real-world images, unlike the synthetic data, do not have explicitly defined Semantic Space and Covariate Space, we indirectly control the range of the Semantic Space through the selection of ID training data. Notably, ImageNet-1K Deng et al. (2009) contains a

Table 3: AUROC (%) of post-hoc OOD detection methods in the image-level experiments

| Training Setup | breed-separated | breed-aggregated | |
|---|---|---|---|
| Testing Protocol | OOD-breed | OOD-breed | OOD-object |
| MSP Hendrycks & Gimpel (2017) | 73.2 ± 0.7 | 50.8 ± 1.3 | 82.4 ± 0.9 |
| ODIN Liang et al. (2018) | 74.5 ± 1.0 | 52.3 ± 1.5 | 81.6 ± 0.4 |
| EBO Liu et al. (2020) | 74.9 ± 1.5 | 51.5 ± 1.6 | 81.5 ± 0.4 |
| GradNorm Huang et al. (2021) | 61.0 ± 1.0 | 52.3 ± 1.8 | 66.7 ± 2.1 |
| RMDS Ren et al. (2021) | 75.4 ± 0.4 | 48.4 ± 2.8 | 81.1 ± 0.5 |
| KNN Sun et al. (2022) | 74.1 ± 0.8 | 50.0 ± 1.5 | 77.4 ± 1.3 |
| DICE Sun & Li (2022) | 71.1 ± 1.7 | 52.2 ± 0.4 | 79.2 ± 1.4 |
| ASH Djurisic et al. (2023) | 73.8 ± 1.0 | 52.2 ± 0.6 | 80.1 ± 1.4 |
| Relation Kim et al. (2023) | 71.7 ± 1.5 | 50.6 ± 1.7 | 80.4 ± 0.6 |
| SCALE Xu et al. (2024) | 70.6 ± 0.5 | 52.2 ± 1.0 | 77.1 ± 1.2 |

substantial number of dog-related classes (126 classes representing various dog breeds). Thus, we utilize the classes within this "dog subset" to manage the Semantic Space of the ID distribution, as illustrated in Fig. 1.

Specifically, in the "breed-separated" setup, we randomly select 100 classes from the "dog subset" as ID data for training. As a comparison, in the "breed-aggregated" setup, we aggregate the previously selected 100 dog classes into a single class and then randomly select 99 additional non-dog classes from ImageNet-1K, resulting in a total of 100 classes for training. As analyzed at the end of Sec. 2, the Semantic Spaces obtained under these two setups are different.

During testing, in the "OOD-breed" protocol, we use the test set of the 100 selected dog classes as ID data and the test set of the remaining 26 dog classes as OOD data. This testing protocol evaluates the OOD detection model's ability to identify novel dog classes under the two training setups. In the "OOD-object" protocol, the test set of the selected 99 non-dog classes serves as ID data, and 20 other non-dog classes from ImageNet-1K are randomly selected as OOD data. This protocol is designed to assess the Semantic Space learned under the 'breed-aggregated' training setup.

We employ a classifier with the ResNet-18 He et al. (2016) as the backbone to verify whether our theoretical analysis holds when using a deep neural network-based nonlinear classifier.

**Performance of OOD detection methods.** The performances of existing post-hoc OOD detection methods on our training setups and testing protocols are summarized in Table 3. To ensure that the results are not influenced by the random selection of classes, we use three different random seeds to sample the classes for each experiment and report both the mean and variance of the outcomes.

When the training setup is "breed-separated", most OOD detection methods achieve an AUROC of over 70% on the "OOD-breed" testing protocol. This indicates that the methods are capable of distinguishing the novel dog classes from the ID dog classes and accurately identifying them as OOD samples. In contrast, when the training setup is "breed-aggregated", the AUROCs of all OOD detection methods on the "OOD-breed" testing protocol drop to around 50%, indicating that these methods struggle to differentiate between OOD and ID dog classes. The experimental results align with our hypothesis: post-hoc OOD detection methods are effective at detecting shifts in the Semantic Space but fail to recognize shifts in the Covariate Space. This confirms that our theoretical analysis holds within high-dimensional image-based input spaces and when nonlinear classifiers are applied.

Finally, the results from the "OOD-object" testing protocol, where the AUROCs of OOD detection methods approach 80%, indicate that the Semantic Space learned by the model in the "breed-aggregated" setup likely represents a high-level semantic feature space. The model can only distinguish between broad categories such as "dog" vs. "ice cream", rather than finer breed-level distinctions like "chihuahua" vs. "beagle".

**Distributions of confidence values.** We present the distribution of confidence values obtained by EBO Liu et al. (2020) across different training setups for ID and OOD samples in the "OOD-breed" testing protocol, as shown in Fig. 3. Under the "breed-separated" training setup, there is a significant difference in the distribution of confidence values between novel dog classes and ID dog classes, allowing the model to effectively distinguish between them. In contrast, when the training setup is "breed-aggregated", the distributions of confidence values for ID and OOD dog classes are nearly identical, further supporting the conclusion of Theorem 1.

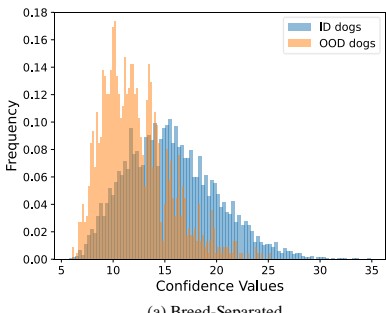 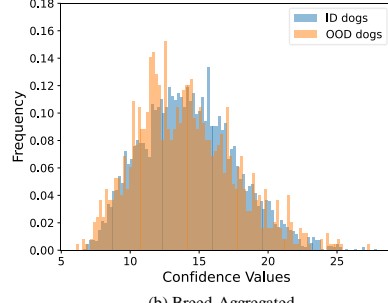

(a) Breed-Separated  (b) Breed-Aggregated

Figure 3: The distributions of EBO's Liu et al. (2020) confidence output under two training setups. In the "breed-separated" training setup, the confidence distributions for ID and OOD dogs show a significant difference, while in the "breed-aggregated" training setup, the two confidence distributions are almost overlapping.

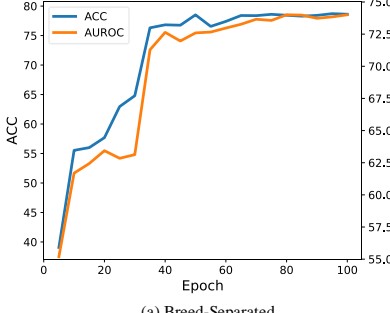 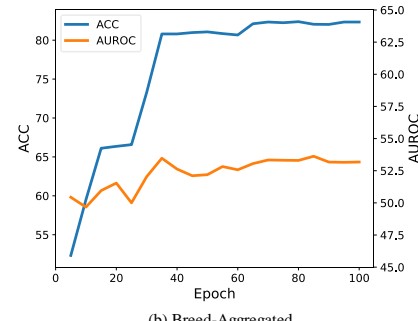

(a) Breed-Separated  (b) Breed-Aggregated

Figure 4: The ID classification ACC and the OOD detection AUROC of EBO during the training process. The OOD detection performance rises along with the ID classification performance in the "breed-separated" training setup. However, the OOD detection AUROC remains around 50% under the "breed-aggregated" training setup.

**Variation of performance during training.** We also investigate the variation in OOD detection performance of EBO Liu et al. (2020) throughout the training process, as illustrated in Fig. 4. In the "breed-separated" training setup, both the classification accuracy for ID classes and the AUROC for OOD detection increase in sync as training progresses. This indicates that the model is gradually learning the "breed-specific feature", improving both its classification performance and OOD detection capability. However, when the training setup is "breed-aggregated", the AUROC for OOD detection remains unchanged regardless of the number of training epochs. This suggests that the model fails to capture the "breed-specific feature" during the training, as the absence of the feature hinders its ability to identify novel dog classes.

**Feature visualization.** Finally, we visualize the features of all dog classes learned by the model in both the "breed-separated" and "breed-aggregated" setups using t-SNE Van der Maaten & Hinton (2008), as shown in Fig. 5. In the "breed-separated" setup, features of different ID dog classes are well-separated, with most OOD dog classes positioned in the gap between the ID features. This allows the post-hoc OOD detection methods to effectively identify these "outliers" as OOD data. However, in the "breed-aggregated" setup, the features of all dog classes cluster closely together with no clear separation. This lack of differentiation makes it intractable for the post-hoc methods to detect novel dog classes as OOD samples.

## 5 RELATED WORK

**OOD Detection Method.** Existing OOD detection methods can be grouped into three main categories. The majority of works focus on post-hoc methods Hendrycks & Gimpel (2017); Liang et al.

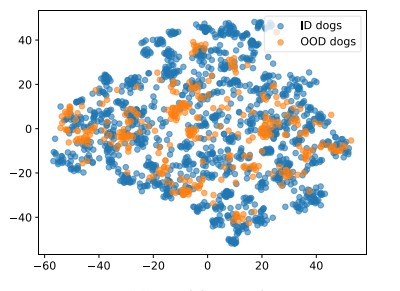 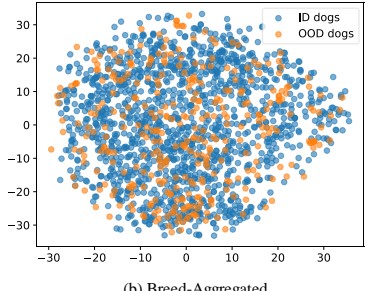

(a) Breed-Separated                    (b) Breed-Aggregated

Figure 5: The t-SNE visualization of the features of ID and OOD dogs extracted by the ResNet-18 trained under "breed-separated" and "breed-aggregated" setups. The separability between OOD and ID features in the "breed-separated" setup is significantly higher than in the "breed-aggregated" setup.

(2018); Lee et al. (2018); Liu et al. (2020); Huang et al. (2021); Sun et al. (2022); Sun & Li (2022); Song et al. (2022); Djurisic et al. (2023); Kim et al. (2023); Xu et al. (2024), which leave the original model training process unchanged and instead utilize the features or prediction outputs of the test data to assess whether a sample is OOD. Another approach involves training-based methods Hsu et al. (2020); Du et al. (2022); Ming et al. (2023); Lu et al. (2024), which alter the model architecture or training process to improve the model's ability to predict OOD classes. A third approach incorporates external OOD data during training Hendrycks et al. (2019); Yu & Aizawa (2019); Yang et al. (2021), helping the model distinguish between ID and OOD data throughout the training phase. Due to the variability in training modifications within the latter two categories, making unified analysis challenging, this paper focuses on the post-hoc OOD detection methods.

**OOD Detection Theory.** Compared to the research on OOD detection methods, fewer works study the theory of OOD detection. Several works on OOD detection theory aim to provide guarantees or bounds for the OOD detection task. One study provides theoretical guarantees for their proposed OOD detection method under Gaussian Mixtures Morteza & Li (2022). The theory in Fang et al. (2022) focuses on the PAC learnability of OOD detection in different scenarios. Another line of research explores factors that can enhance OOD detection performance. The additional unlabeled data is proved to be beneficial for the OOD detection task in Du et al. (2024a). Another work shows that ID labels can also significantly improve OOD detection performance Du et al. (2024b). In contrast to the aforementioned works, we focus more on the flaws in the current definition of OOD detection settings and address these issues by offering a more precise definition. An interesting study explores the issues with using deep generative models for OOD detection Zhang et al. (2021). In comparison, we study the challenges present in post-hoc methods based on classification models.

## 6 CONCLUSION AND DISCUSSION

In this paper, we identify flaws in the existing OOD detection settings and introduce a more precise definition of the Semantic Space and the corresponding Covariate Space for the OOD detection task. Based on this definition, we theoretically analyze cases where OOD detection becomes intractable for post-hoc methods. To address the issue, we further propose the "Tractable OOD" setting, which ensures the distinguishability between OOD and ID distributions during the OOD testing process. Finally, we conduct extensive experiments that validate our theoretical analysis.

**Limitation.** The theoretical analysis is conducted in low-dimensional spaces for simplification, under the assumption of linear separability between Gaussian-like classes. Although experimental results confirm the validity of our theorem with high-dimensional image inputs and nonlinear classifiers, there remains a lack of theoretical proof for more general training scenarios. Future work can explore the definitions of Semantic Space and Covariate Space and theoretically study their impact on OOD detection tasks without imposing restrictions on the input space and classifier.

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

# A PROOFS

## A.1 PROOF OF PROPOSITION 1

For any two different representative feature vectors $\boldsymbol{\mu}_i, \boldsymbol{\mu}_j (i, j \in \{1, ..., k\}, i < j)$ from ID distributions, we get their decomposition within the Semantic Space and the Covariate Space as follows

$$\begin{aligned} \boldsymbol{\mu}_i = \boldsymbol{s}_i + \boldsymbol{c}_i, \boldsymbol{s}_i \in \mathcal{S}, \boldsymbol{c}_i \in \mathcal{C}, \\ \boldsymbol{\mu}_j = \boldsymbol{s}_j + \boldsymbol{c}_j, \boldsymbol{s}_j \in \mathcal{S}, \boldsymbol{c}_j \in \mathcal{C}. \end{aligned} \tag{13}$$

By subtracting the two equations and making some adjustments, we obtain

$$(\boldsymbol{\mu}_i - \boldsymbol{\mu}_j) - (\boldsymbol{s}_i - \boldsymbol{s}_j) = (\boldsymbol{c}_i - \boldsymbol{c}_j). \tag{14}$$

According to our definition of the Semantic Space that $\mathcal{S} = span(\{\boldsymbol{\mu}_1 - \boldsymbol{\mu}_2, ..., \boldsymbol{\mu}_k - \boldsymbol{\mu}_{k-1}\})$, we know that

$$(\boldsymbol{\mu}_i - \boldsymbol{\mu}_{i+1}) + ... + (\boldsymbol{\mu}_{j-1} - \boldsymbol{\mu}_j) = \boldsymbol{\mu}_i - \boldsymbol{\mu}_j \in \mathcal{S}. \tag{15}$$

Since $\boldsymbol{s}_i, \boldsymbol{s}_j \in \mathcal{S}$, it implies that

$$\boldsymbol{s}_i - \boldsymbol{s}_j \in \mathcal{S}. \tag{16}$$

Substituting Eq. (15) and Eq. (16) into Eq. (14), we know that

$$\boldsymbol{c}_i - \boldsymbol{c}_j \in \mathcal{S}. \tag{17}$$

However, since $\boldsymbol{c}_i, \boldsymbol{c}_j \in \mathcal{C}$,

$$\boldsymbol{c}_i - \boldsymbol{c}_j \in \mathcal{C}. \tag{18}$$

According to the definition of direct sum decomposition $\mathcal{X} = \mathcal{S} \oplus \mathcal{C}$,

$$\mathcal{S} \cap \mathcal{C} = \boldsymbol{0}. \tag{19}$$

We can finally obtain

$$\boldsymbol{c}_i - \boldsymbol{c}_j = \boldsymbol{0}, \tag{20}$$

which implies

$$\boldsymbol{c}_i = \boldsymbol{c}_j \equiv \boldsymbol{c}_{const}. \tag{21}$$

We have completed this proof.

## A.2 NECESSARY LEMMA FOR PROPOSITION 2

To prove Proposition 2, we first propose the following lemma to ensure that the training process of the linear classifier remains unaffected by an orthogonal transformation.

**Lemma 1.** *Given an orthogonal matrix $\boldsymbol{Q}$, training a linear classifier $f_1(\cdot)$ with a weight matrix $\boldsymbol{W}$ on the ID distribution $(\boldsymbol{x}, y) \sim \mathcal{P}_I$ is equivalent to training a linear classifier $f_2(\cdot)$ with another weight matrix $\tilde{\boldsymbol{W}} = \boldsymbol{W} \cdot \boldsymbol{Q}^\top$ on the orthogonally transformed distribution $(\boldsymbol{x}, y) \sim \tilde{\mathcal{P}}_I$. The orthogonally transformed distribution $\tilde{\mathcal{P}}_I$ denotes the distribution of $(\boldsymbol{Q}\boldsymbol{x}, y)$ when $(\boldsymbol{x}, y)$ is sampled from the original ID distribution $(\boldsymbol{x}, y) \sim \mathcal{P}_I$.*

**Proof of Lemma 1.** We only need to prove that if, at a certain training step $n$, the weight matrices of the two linear classifiers satisfy condition $\boldsymbol{W}_n = \tilde{\boldsymbol{W}}_n \cdot \boldsymbol{Q}$, and after their respective next optimization steps, the two weight matrices still satisfy $\boldsymbol{W}_{n+1} = \tilde{\boldsymbol{W}}_{n+1} \cdot \boldsymbol{Q}$, then it can be demonstrated that the two training setups are equivalent.

We can find that the outputs of the two models $f_1(\boldsymbol{x}_1)$ and $f_2(\boldsymbol{x}_2)$ are consistent when their inputs follows the relation $\boldsymbol{x}_2 = \boldsymbol{Q}\boldsymbol{x}_1$

$$\begin{aligned} f_1(\boldsymbol{x}_1) &= softmax(\boldsymbol{W}\boldsymbol{x}_1) \\ &= softmax((\tilde{\boldsymbol{W}} \cdot \boldsymbol{Q})(\boldsymbol{Q}^\top \boldsymbol{x}_2)) \\ &= softmax(\tilde{\boldsymbol{W}}\boldsymbol{x}_2) \\ &= f_2(\boldsymbol{x}_2). \end{aligned} \tag{22}$$

Since the training loss consists of cross-entropy loss and L2 regularization loss as follows (In the proofs of this lemma and subsequent Proposition 2, we assume that the training loss consists solely of these two components, consistent with the classification training in most works.)

$$
\begin{aligned}
L_1 &= -\mathbb{E}_{\boldsymbol{x},y\sim\mathcal{P}_I}[onehot(y)^\top \cdot log(f_1(\boldsymbol{x}))] + \frac{\lambda}{2}\|\boldsymbol{W}\|_2^2, \\
L_2 &= -\mathbb{E}_{\boldsymbol{x},y\sim\tilde{\mathcal{P}}_I}[onehot(y)^\top \cdot log(f_2(\boldsymbol{x}))] + \frac{\lambda}{2}\|\tilde{\boldsymbol{W}}\|_2^2,
\end{aligned}
\tag{23}
$$

where $L_1$ and $L_2$ denote the training loss of the two setups, and $onehot(y)$ refers to the one-hot vector of the label $y$.

We can then bridge the gradient of the two setups.

$$
\begin{aligned}
\frac{\partial L_1}{\partial \boldsymbol{W}} &= \mathbb{E}_{\boldsymbol{x},y\sim\mathcal{P}_I}[(f_1(\boldsymbol{x}) - onehot(y))\boldsymbol{x}^\top] + \lambda\boldsymbol{W} \\
&= \mathbb{E}_{\boldsymbol{x},y\sim\mathcal{P}_I}[(f_2(\boldsymbol{Q}\boldsymbol{x}) - onehot(y))\boldsymbol{x}^\top] + \lambda\tilde{\boldsymbol{W}}\cdot\boldsymbol{Q} \\
&= \mathbb{E}_{\boldsymbol{x},y\sim\mathcal{P}_I}[(f_2(\boldsymbol{Q}\boldsymbol{x}) - onehot(y))\boldsymbol{x}^\top\boldsymbol{Q}^\top]\cdot\boldsymbol{Q} + \lambda\tilde{\boldsymbol{W}}\cdot\boldsymbol{Q} \\
&= \mathbb{E}_{\boldsymbol{x},y\sim\mathcal{P}_I}[(f_2(\boldsymbol{Q}\boldsymbol{x}) - onehot(y))(\boldsymbol{Q}\boldsymbol{x})^\top]\cdot\boldsymbol{Q} + \lambda\tilde{\boldsymbol{W}}\cdot\boldsymbol{Q} \\
&= \mathbb{E}_{\boldsymbol{x},y\sim\tilde{\mathcal{P}}_I}[(f_2(\boldsymbol{x}) - onehot(y))\boldsymbol{x}^\top]\cdot\boldsymbol{Q} + \lambda\tilde{\boldsymbol{W}}\cdot\boldsymbol{Q} \\
&= \frac{\partial L_2}{\partial \tilde{\boldsymbol{W}}}\cdot\boldsymbol{Q}.
\end{aligned}
\tag{24}
$$

After a single optimization step, the gradient decent process of the two classifiers are

$$
\begin{aligned}
\boldsymbol{W}_{n+1} &= \boldsymbol{W}_n - \eta\frac{\partial L_1}{\partial \boldsymbol{W}_n}, \\
\tilde{\boldsymbol{W}}_{n+1} &= \tilde{\boldsymbol{W}}_n - \eta\frac{\partial L_2}{\partial \tilde{\boldsymbol{W}}_n},
\end{aligned}
\tag{25}
$$

where $\eta$ is the learning rate.

We can then use Eq. (24) to obtain

$$
\begin{aligned}
\boldsymbol{W}_{n+1} &= \boldsymbol{W}_n - \eta\frac{\partial L_1}{\partial \boldsymbol{W}_n}, \\
&= \tilde{\boldsymbol{W}}_n\cdot\boldsymbol{Q} - \eta\frac{\partial L_2}{\partial \tilde{\boldsymbol{W}}_n}\cdot\boldsymbol{Q}, \\
&= (\tilde{\boldsymbol{W}}_n - \eta\frac{\partial L_2}{\partial \tilde{\boldsymbol{W}}_n})\cdot\boldsymbol{Q}, \\
&= \tilde{\boldsymbol{W}}_{n+1}\cdot\boldsymbol{Q},
\end{aligned}
\tag{26}
$$

We have completed this proof.

### A.3 PROOF OF PROPOSITION 2

With Lemma 1 established, we can construct an orthogonal matrix $\boldsymbol{Q}$ to obtain the orthogonally transformed distribution $\tilde{\mathcal{P}}_I$, ensuring the Semantic Space and the Covariate Space are decoupled. The construction process of the orthogonal matrix $\boldsymbol{Q}$ is as follows.

We first apply the Gram-Schmidt orthogonalization to obtain a set of orthonormal basis vectors for the Semantic Space $\mathcal{S}$

$$
\{\boldsymbol{q}_1, ..., \boldsymbol{q}_r\} = GS(\{\boldsymbol{\mu}_1 - \boldsymbol{\mu}_2, ..., \boldsymbol{\mu}_k - \boldsymbol{\mu}_{k-1}\}),
\tag{27}
$$

where $GS(\cdot)$ represents the Gram-Schmidt orthogonalization. Note that the input vector set $\{\boldsymbol{\mu}_1 - \boldsymbol{\mu}_2, ..., \boldsymbol{\mu}_k - \boldsymbol{\mu}_{k-1}\}$ may not be linearly independent, so during the Gram-Schmidt orthogonalization process, zero vectors may appear. We skip these vectors and only retain the non-zero vectors as the

basis, denoted as $\{\boldsymbol{q}_1, ..., \boldsymbol{q}_r\}$, $r \leq k - 1$. Therefore, the number $r$ of basis vectors corresponds to the rank of the Semantic Space.

We then extend this orthonormal basis to $d$ dimensions, yielding a set of orthonormal basis vectors for the input space $\mathcal{X}$.

$$\{\boldsymbol{q}_1, ..., \boldsymbol{q}_r, \boldsymbol{q}_{r+1}, ..., \boldsymbol{q}_d\} = GS(\{\boldsymbol{q}_1, ..., \boldsymbol{q}_r, \boldsymbol{e}_1, ..., \boldsymbol{e}_d\}),$$
$$\boldsymbol{e}_i = [0, ..., 0, 1, 0, ..., 0]^\top, \tag{28}$$

where $\boldsymbol{e}_i$ is a standard basis vector with the $i^{th}$ component is 1 and all other components are 0. We concatenate the Semantic Space basis vectors $\{\boldsymbol{q}_1, ..., \boldsymbol{q}_r\}$ with the standard basis vectors $\{\boldsymbol{e}_1, ..., \boldsymbol{e}_d\}$ before the Gram-Schmidt orthogonalization, ensuring the completeness of the final $d$-dimensional basis. Since the input space $\mathcal{X}$ is decomposed into the direct sum of subspaces $\mathcal{S}$ and $\mathcal{C}$, it is easy to see that $\{\boldsymbol{q}_{r+1}, ..., \boldsymbol{q}_d\}$ forms a set of orthonormal basis vectors for the Covariate Space $\mathcal{C}$.

At this point, we obtain the orthonormal bases for both the Semantic Space and the Covariate Space

$$span(\{\boldsymbol{q}_1, ..., \boldsymbol{q}_r\}) = span(\{\boldsymbol{\mu}_1 - \boldsymbol{\mu}_2, ..., \boldsymbol{\mu}_k - \boldsymbol{\mu}_{k-1}\}) = \mathcal{S},$$
$$span(\{\boldsymbol{q}_{r+1}, ..., \boldsymbol{q}_d\}) = \mathcal{C}. \tag{29}$$

Thus, each representative feature vector can be expressed in terms of these basis vectors

$$\boldsymbol{\mu}_i = \boldsymbol{s}_i + \boldsymbol{c}_{const}, \boldsymbol{s}_i \in \mathcal{S}, \boldsymbol{c}_{const} \in \mathcal{C},$$
$$\boldsymbol{s}_i = s_{i1}\boldsymbol{q}_1 + ... + s_{ir}\boldsymbol{q}_r, \tag{30}$$
$$\boldsymbol{c}_{const} = c_{r+1}\boldsymbol{q}_{r+1} + ... + c_d\boldsymbol{q}_d.$$

Based on the basis vectors, we can construct an orthogonal matrix $\boldsymbol{Q}$ to decompose the Semantic Space and the Covariate Space

$$\boldsymbol{Q} = [\boldsymbol{q}_1, ..., \boldsymbol{q}_r, \boldsymbol{q}_{r+1}, ..., \boldsymbol{q}_d]^\top. \tag{31}$$

At this point, we complete the construction of the orthogonal matrix $\boldsymbol{Q}$, and we obtain the new input distribution $(\boldsymbol{x}, y) \sim \tilde{\mathcal{P}}_I$. After the orthogonal transformation, the ID representative feature vectors become

$$\tilde{\boldsymbol{\mu}}_i = Q \cdot \boldsymbol{\mu}_i = [s_{i1}, ..., s_{ir}, c_{r+1}, ..., c_d]^\top, i \in \{1, ..., k\}. \tag{32}$$

Since a Gaussian distribution remains Gaussian after an orthogonal transformation, the input distribution of each ID class can now be expressed as follows

$$\mathcal{N}(\tilde{\boldsymbol{\mu}}_i, \boldsymbol{I}), i \in \{1, ..., k\}. \tag{33}$$

As a result, after applying Lemma 1, the proof of Proposition 2 reduces to studying the training process of a new linear classifier with a simplified weight matrix $\tilde{\boldsymbol{W}} = \boldsymbol{W} \cdot \boldsymbol{Q}^\top$ within the orthogonally transformed distribution $\tilde{\mathcal{P}}_I$, where the ID distribution follows

$$\tilde{\mathcal{P}}_I | y_i = (\mathcal{N}(\tilde{\boldsymbol{\mu}}_i, \boldsymbol{I}), y_i), i \in \{1, ..., k\}, \tilde{\boldsymbol{\mu}}_i = [s_{i1}, ..., s_{ir}, c_{r+1}, ..., c_d]^\top. \tag{34}$$

The last $d - r$ dimensions of the representative feature vectors are constants and do not vary with the input label.

The output of the new linear classifier is now as follows

$$f(\boldsymbol{x}) = softmax(\tilde{\boldsymbol{W}}\boldsymbol{x}) = [\tilde{p}_1(\boldsymbol{x}), ..., \tilde{p}_k(\boldsymbol{x})]^\top,$$
$$\tilde{p}_i(\boldsymbol{x}) = \frac{\exp(\tilde{\boldsymbol{W}}_{i,:}\boldsymbol{x})}{\sum_{j=1}^k \exp(\tilde{\boldsymbol{W}}_{j,:}\boldsymbol{x})}, i \in \{1, ..., k\}. \tag{35}$$

Considering the training process, the total loss is set as

$$Loss = -\mathbb{E}_{\boldsymbol{x},y}[\sum_{i=1}^k \mathbb{I}\{y = i\}log(\tilde{p}_i(\boldsymbol{x}))] + \frac{\lambda}{2}\|\tilde{\boldsymbol{W}}\|_2^2, \tag{36}$$

where the first term is the cross-entropy loss, and the second term is the L2 regularization loss. $\mathbb{I}\{y = i\}$ represents the indicator, which equals 1 when $y = i$, and 0 otherwise. The partial derivative of the loss function with respect to the classifier's weight of class $i$ on a specific input dimension $j$ is then computed as follows

$$\frac{\partial Loss}{\partial \tilde{\boldsymbol{W}}_{ij}} = \mathbb{E}_{\boldsymbol{x},y}[(\tilde{p}_i(\boldsymbol{x}) - \mathbb{I}\{y = i\})\boldsymbol{x}_j] + \lambda\tilde{\boldsymbol{W}}_{ij}. \tag{37}$$

Now we examine the last $d - r$ columns of the simplified matrix $\tilde{\boldsymbol{W}}$. Since when $j > r$, the value of $\boldsymbol{x}_j$ is independent with the input label $y$. We can derive the first term of Eq. (37) as follows

$$\begin{aligned}
\mathbb{E}_{\boldsymbol{x},y}[(\tilde{p}_i(\boldsymbol{x}) - \mathbb{I}\{y = i\})\boldsymbol{x}_j] &= \mathbb{E}_{\boldsymbol{x},y}[\tilde{p}_i(\boldsymbol{x})\boldsymbol{x}_j] - \mathbb{E}_{\boldsymbol{x},y}[\mathbb{I}\{y = i\}\boldsymbol{x}_j] \\
&= \mathbb{E}_{\boldsymbol{x},y}[\tilde{p}_i(\boldsymbol{x})\boldsymbol{x}_j] - \mathbb{E}_y[\mathbb{I}\{y = i\}]\mathbb{E}_{\boldsymbol{x}}[\boldsymbol{x}_j] \\
&= \mathbb{E}_{\boldsymbol{x},y}[\tilde{p}_i(\boldsymbol{x})\boldsymbol{x}_j] - \frac{1}{k} \cdot c_j \\
&= \mathbb{E}_{\boldsymbol{x},y}[\tilde{p}_i(\boldsymbol{x})]\mathbb{E}_{\boldsymbol{x},y}[\boldsymbol{x}_j] + Cov(\tilde{p}_i(\boldsymbol{x}), \boldsymbol{x}_j) - \frac{c_j}{k}
\end{aligned} \tag{38}$$

Applying Assumption 1, we obtain

$$\begin{aligned}
\mathbb{E}_{\boldsymbol{x},y}[(\tilde{p}_i(\boldsymbol{x}) - \mathbb{I}\{y = i\})\boldsymbol{x}_j] &= \mathbb{E}_{\boldsymbol{x},y}[\tilde{p}_i(\boldsymbol{x})]\mathbb{E}_{\boldsymbol{x},y}[\boldsymbol{x}_j] + Cov(\tilde{p}_i(\boldsymbol{x}), \boldsymbol{x}_j) - \frac{c_j}{k} \\
&= \frac{1}{k} \cdot c_j + Cov(\tilde{p}_i(\boldsymbol{x}), \boldsymbol{x}_j) - \frac{c_j}{k} \\
&= Cov(\tilde{p}_i(\boldsymbol{x}), \boldsymbol{x}_j)
\end{aligned} \tag{39}$$

Substituting this result back into Eq. (37), the expression for the partial derivative becomes

$$\frac{\partial Loss}{\partial \tilde{\boldsymbol{W}}_{ij}} = Cov(\tilde{p}_i(\boldsymbol{x}), \boldsymbol{x}_j) + \lambda\tilde{\boldsymbol{W}}_{ij}. \tag{40}$$

We multiply the partial derivative by the weight $\tilde{\boldsymbol{W}}_{ij}$ itself to analyze the direction of the weight update

$$\tilde{\boldsymbol{W}}_{ij} \cdot \frac{\partial Loss}{\partial \tilde{\boldsymbol{W}}_{ij}} = \tilde{\boldsymbol{W}}_{ij} \cdot Cov(\tilde{p}_i(\boldsymbol{x}), \boldsymbol{x}_j) + \lambda\tilde{\boldsymbol{W}}_{ij}^2. \tag{41}$$

After applying Assumption 2,

$$\tilde{\boldsymbol{W}}_{ij} \cdot \frac{\partial Loss}{\partial \tilde{\boldsymbol{W}}_{ij}} \geq \lambda\tilde{\boldsymbol{W}}_{ij}^2 \geq 0. \tag{42}$$

The derivation shows that for a given weight $\tilde{\boldsymbol{W}}_{ij}$ of class $i$ on the $j^{th}$ input dimension, $j > r$, the partial derivative of the loss function with respect to that weight has the same sign as the weight itself. This means that the absolute value of the weight will continuously decrease throughout the training process due to gradient descent optimization.

Next, we will further prove that $\tilde{\boldsymbol{W}}_{ij}$ converges to 0 when $j > r$. Considering the gradient descent process

$$\tilde{\boldsymbol{W}}_{n+1,ij} = \tilde{\boldsymbol{W}}_{n,ij} - \eta\frac{\partial Loss}{\partial \tilde{\boldsymbol{W}}_{n,ij}}, \tag{43}$$

where $n$ denotes the training steps, $\tilde{\boldsymbol{W}}_{n,ij}$ is the weight in training step $n$, and $\eta$ refers to the learning rate. In the training setup, the learning rate $\eta$ is set to a small value to ensure the convergence of the training process. Therefore, we assume in the training process,

$$|\tilde{\boldsymbol{W}}_{n,ij}| \geq |\eta\frac{\partial Loss}{\partial \tilde{\boldsymbol{W}}_{n,ij}}|. \tag{44}$$

Combined with Eq. (42), we obtain

$$\tilde{\boldsymbol{W}}_{n,ij} \cdot \eta\frac{\partial Loss}{\partial \tilde{\boldsymbol{W}}_{n,ij}} \geq (\eta\frac{\partial Loss}{\partial \tilde{\boldsymbol{W}}_{n,ij}})^2. \tag{45}$$

Squaring both sides of Eq. (43) and substituting Eq. (42) and Eq. (45),

$$
\begin{aligned}
\tilde{\boldsymbol{W}}_{n+1,ij}^2 &= \tilde{\boldsymbol{W}}_{n,ij}^2 + (\eta \frac{\partial Loss}{\partial \tilde{\boldsymbol{W}}_{n,ij}})^2 - 2\tilde{\boldsymbol{W}}_{n,ij} \cdot \eta \frac{\partial Loss}{\partial \tilde{\boldsymbol{W}}_{n,ij}} \\
&= \tilde{\boldsymbol{W}}_{n,ij}^2 + ((\eta \frac{\partial Loss}{\partial \tilde{\boldsymbol{W}}_{n,ij}})^2 - \tilde{\boldsymbol{W}}_{n,ij} \cdot \eta \frac{\partial Loss}{\partial \tilde{\boldsymbol{W}}_{n,ij}}) - \tilde{\boldsymbol{W}}_{n,ij} \cdot \eta \frac{\partial Loss}{\partial \tilde{\boldsymbol{W}}_{n,ij}} \\
&\leq \tilde{\boldsymbol{W}}_{n,ij}^2 - \tilde{\boldsymbol{W}}_{n,ij} \cdot \eta \frac{\partial Loss}{\partial \tilde{\boldsymbol{W}}_{n,ij}} \\
&= \tilde{\boldsymbol{W}}_{n,ij}^2 - \eta \cdot (\tilde{\boldsymbol{W}}_{n,ij} \cdot \frac{\partial Loss}{\partial \tilde{\boldsymbol{W}}_{n,ij}}) \\
&\leq \tilde{\boldsymbol{W}}_{n,ij}^2 - \eta \cdot \lambda \tilde{\boldsymbol{W}}_{ij}^2 \\
&= (1 - \eta\lambda)\tilde{\boldsymbol{W}}_{n,ij}^2
\end{aligned}
\tag{46}
$$

Thus, the final convergence result is

$$
|\tilde{\boldsymbol{W}}_{n+1,ij}| \leq (1-\eta\lambda)^{\frac{1}{2}} |\tilde{\boldsymbol{W}}_{n,ij}| \leq (1-\eta\lambda)^{\frac{n}{2}} |\tilde{\boldsymbol{W}}_{1,ij}| \xrightarrow{n \to \infty} 0.
\tag{47}
$$

The equation holds because $\eta$ and $\lambda$ are set to small positive values, which ensures $0 < (1-\eta\lambda) < 1$.

The result implies that once the training has converged, $\tilde{\boldsymbol{W}}_{ij}$ converges to 0 when $j > r$, which means

$$
\tilde{\boldsymbol{W}}_{:,r+1} = ... = \tilde{\boldsymbol{W}}_{:,d} = \boldsymbol{0}.
\tag{48}
$$

We have completed this proof.

### A.4 PROOF OF THEOREM 1

Since the representative feature vectors of the two classes of data $\mathcal{N}(\boldsymbol{\mu}_a, \boldsymbol{I})$ and $\mathcal{N}(\boldsymbol{\mu}_b, \boldsymbol{I})$ are identical in the Semantic Space $\mathcal{S}$, we assume the decomposition of the two representative feature vectors is as follows

$$
\begin{aligned}
\boldsymbol{\mu}_a &= \boldsymbol{s} + \boldsymbol{c}_a, \\
\boldsymbol{\mu}_b &= \boldsymbol{s} + \boldsymbol{c}_b, \\
\boldsymbol{s} &\in \mathcal{S}, \quad \boldsymbol{c}_a, \boldsymbol{c}_b \in \mathcal{C}.
\end{aligned}
\tag{49}
$$

Using the same method from the proof of Proposition 2 to construct the orthogonal basis vectors and the orthogonal matrix, we can express the two representative feature vectors in terms of the basis

$$
\begin{aligned}
\boldsymbol{\mu}_a &= s_1 \boldsymbol{q}_1 + ... + s_r \boldsymbol{q}_r + c_{a,r+1} \boldsymbol{q}_{r+1} + ... + c_{a,d} \boldsymbol{q}_d, \\
\boldsymbol{\mu}_b &= s_1 \boldsymbol{q}_1 + ... + s_r \boldsymbol{q}_r + c_{b,r+1} \boldsymbol{q}_{r+1} + ... + c_{b,d} \boldsymbol{q}_d.
\end{aligned}
\tag{50}
$$

According to the linear transformation of Gaussians, we can derive the output distribution $\mathcal{P}_a$ of the input distribution $\mathcal{N}(\boldsymbol{\mu}_a, \boldsymbol{I})$ after being transformed by the weight matrix $\boldsymbol{W}$

$$
\begin{aligned}
\boldsymbol{o}_a &= \boldsymbol{W} \boldsymbol{x}_a, \\
\boldsymbol{x} &\sim \mathcal{N}(\boldsymbol{\mu}_a, \boldsymbol{I}), \\
\boldsymbol{o}_a &\sim \mathcal{P}_a = \mathcal{N}(\boldsymbol{W}\boldsymbol{\mu}_a, \boldsymbol{W}\boldsymbol{I}\boldsymbol{W}^\top).
\end{aligned}
\tag{51}
$$

Based on Proposition 2 that $\boldsymbol{W} = \tilde{\boldsymbol{W}} \cdot \boldsymbol{Q}$, we can conclude

$$
\begin{aligned}
\boldsymbol{W}\boldsymbol{\mu}_a &= \tilde{\boldsymbol{W}} \boldsymbol{Q} \boldsymbol{\mu}_a \\
&= \tilde{\boldsymbol{W}} [\boldsymbol{q}_1, ..., \boldsymbol{q}_r, \boldsymbol{q}_{r+1}, ..., \boldsymbol{q}_d]^\top (s_1 \boldsymbol{q}_1 + ... + s_r \boldsymbol{q}_r + c_{a,r+1} \boldsymbol{q}_{r+1} + ... + c_{a,d} \boldsymbol{q}_d) \\
&= \tilde{\boldsymbol{W}} [s_1, ..., s_r, c_{a,r+1}, ..., c_{a,d}]^\top \\
&= [\tilde{\boldsymbol{W}}_{:,1}, ..., \tilde{\boldsymbol{W}}_{:,r}, 0, ..., 0][s_1, ..., s_r, c_{a,r+1}, ..., c_{a,d}]^\top \\
&= s_1 \tilde{\boldsymbol{W}}_{:,1} + ... + s_r \tilde{\boldsymbol{W}}_{:,r}.
\end{aligned}
\tag{52}
$$

This implies the output distribution of the input distribution $\mathcal{N}(\boldsymbol{\mu}_a, \boldsymbol{I})$ is

$$\boldsymbol{o}_a \sim \mathcal{P}_a = \mathcal{N}(s_1 \tilde{\boldsymbol{W}}_{:,1} + ... + s_r \tilde{\boldsymbol{W}}_{:,r}, \boldsymbol{W}\boldsymbol{I}\boldsymbol{W}^\top). \tag{53}$$

It can be observed that the output distribution is independent of the Covariate Space component $\boldsymbol{c}_a$ of the input's representative feature vector $\boldsymbol{\mu}_a$.

Using the same method, we can obtain the output distribution of the other input distribution $\mathcal{N}(\boldsymbol{\mu}_b, \boldsymbol{I})$

$$\boldsymbol{o}_b \sim \mathcal{P}_b = \mathcal{N}(s_1 \tilde{\boldsymbol{W}}_{:,1} + ... + s_r \tilde{\boldsymbol{W}}_{:,r}, \boldsymbol{W}\boldsymbol{I}\boldsymbol{W}^\top), \tag{54}$$

which is the same as the output of $\mathcal{N}(\boldsymbol{\mu}_a, \boldsymbol{I})$. It implies

$$KL(\mathcal{P}_a || \mathcal{P}_b) = 0. \tag{55}$$

Since the classifier output just includes an additional softmax layer after applying the weight matrix $\boldsymbol{W}$

$$f(\boldsymbol{x}) = softmax(\boldsymbol{W}\boldsymbol{x}) = softmax(\boldsymbol{o}). \tag{56}$$

Thus, the KL divergence between the classifier's output of the two distributions is

$$\begin{aligned} &KL(f(\mathcal{N}(\boldsymbol{\mu}_a, \boldsymbol{I})) || f(\mathcal{N}(\boldsymbol{\mu}_b, \boldsymbol{I}))) \\ =& KL(softmax(\mathcal{P}_a) || softmax(\mathcal{P}_b)) \\ =& 0 \end{aligned} \tag{57}$$

We have completed this proof.

## B   TRAINING DETAILS IN EXPERIMENTS

The specific parameter settings for the experiment "Synthetic Data" are as follows. The value of $\sigma$, which determines the position of the representative feature vectors, is set to 2. For each epoch, 1000 data points are randomly sampled from the four ID Gaussian distributions for training. The SGD optimizer is used for training, with a learning rate of 0.01, weight decay set to 0.01, and momentum set to 0.9. The model is trained for a total of 5000 epochs.

The parameter settings for the experiment "ImageNet Dogs" are as follows. The model is trained using the training set of the selected ID classes from the ImageNet-1K. In the "breed-aggregated" training setup, as dog classes are merged into one, the training data is balanced to ensure that each class has an equal sampling probability. The classifier used is ResNet-18, with the output set to 100 classes. The preprocessing methods for all images follow the standard ImageNet-1K preprocessing. The batch size is set to 128. We train the model using the SGD optimizer with an initial learning rate of 0.1, weight decay of 1e-4, and momentum of 0.9. A StepLR scheduler is used to adjust the learning rate, reducing it by a factor of 0.1 every 30 epochs. The total number of training epochs is set to 100.

## C   VALIDATION OF PROPOSED ASSUMPTIONS.

We investigate the validity of the two proposed assumptions in the "Synthetic Data" experiment. As shown in Fig. 6, the model maintains a balanced prediction for the four ID classes throughout the entire training process, with no bias toward any specific class. This demonstrates the correctness of Assumption 1.

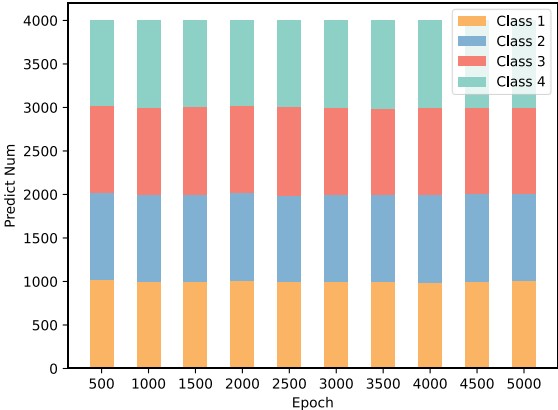

Figure 6: The model's prediction numbers for the four ID classes during the training process.

Additionally, we present the product of the weight $\boldsymbol{W}_{ij}$ and the corresponding covariance $Cov(p_i(\boldsymbol{x}), \boldsymbol{x}_j)$ over the course of training, as shown in Fig. 7. It can be observed that this product is either positive or approaches zero throughout the training process, which validates the correctness of Assumption 2. Notably, when the absolute value of $\boldsymbol{W}_{ij}$ is small, the value of the product $\boldsymbol{W}_{ij} \cdot Cov(p_i(\boldsymbol{x}), \boldsymbol{x}_j)$ tends to be a positive value close to zero. However, due to fluctuations in data sampling, this product may occasionally result in a small negative value during practical training iterations. This is a normal occurrence and does not affect the validity of our assumption.

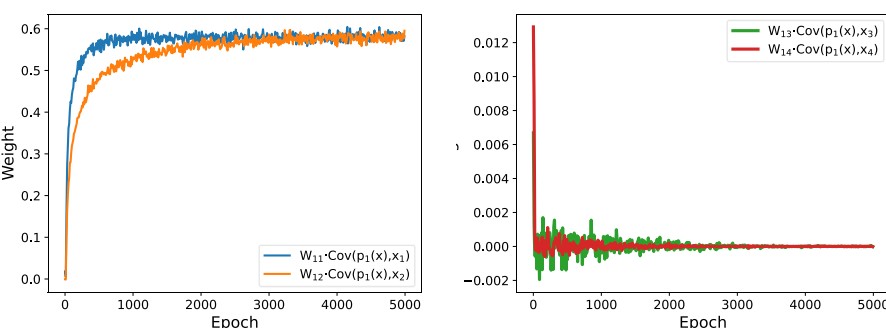

Figure 7: The product of the weight $\boldsymbol{W}_{1j}$ and the corresponding covariance $Cov(p_1(\boldsymbol{x}), \boldsymbol{x}_j)$ during training.

Table 4: AUROC (%) of OOD detection methods on data with orthogonal transformation

| | w/o shift in Semantic Space | | | with shift in Semantic Space | | |
|---|---|---|---|---|---|---|
| $\boldsymbol{s}_o$ | $[\sigma, \sigma, 0, 0]^\top$ | | | $[0, \sigma, 0, 0]^\top$ | | |
| $\boldsymbol{c}_o$ | $[0, 0, \sigma, \sigma]^\top$ | $[0, 0, -\sigma, \sigma]^\top$ | $[0, 0, -\sigma, -\sigma]^\top$ | $[0, 0, \sigma, \sigma]^\top$ | $[0, 0, -\sigma, \sigma]^\top$ | $[0, 0, -\sigma, -\sigma]^\top$ |
| MSP Hendrycks & Gimpel (2017) | 51.1 | 50.7 | 51.5 | 79.3 | 79.7 | 79.5 |
| EBO Liu et al. (2020) | 51.4 | 51.3 | 51.5 | 73.4 | 74.8 | 74.7 |
| GradNorm Huang et al. (2021) | 50.8 | 45.5 | 45.7 | 64.2 | 70.0 | 66.2 |

## D TRAINING ON DATA WITH ORTHOGONAL TRANSFORMATIONS.

We also conduct experiments where the Semantic Space and the Covariate Space are scrambled, which is a more generalized input condition. Specifically, we randomly generate a fixed orthogonal matrix $\boldsymbol{Q}$ and apply it to each input data before training the classifier. The results in Table 4 show that this scrambling does not affect the performance of the OOD detection methods, as the outcomes remain consistent with those shown in Table 1. We further decompose the optimized weight matrix $\boldsymbol{W}$ as shown in Fig. 8, revealing that it can be expressed as the product of a simplified weight matrix $\tilde{\boldsymbol{W}}$ and an orthogonal matrix $\boldsymbol{Q}$. This confirms the validity of our Proposition 2 in the generalized input condition.



Figure 8: The optimized weight matrix $\boldsymbol{W}$ when the Semantic Space and the Covariate Space are scrambled. The weight matrix $\boldsymbol{W}$ can be decomposed into a simplified weight matrix $\tilde{\boldsymbol{W}}$ and an orthogonal matrix $\boldsymbol{Q}$ as demonstrated in our Proposition 2.

