# OpenReview forum: "Semantic or Covariate? A Study on the Intractable Case of Out-of-Distribution Detection"
_ICLR.cc/2025/Conference — ICLR 2025 Conference Withdrawn Submission_

### Official Review · Reviewer_51NN · 2024-10-29

**Soundness:** 1
**Presentation:** 2
**Contribution:** 1
**Rating:** 3
**Confidence:** 3

**Summary:**

This paper focuses on identifying the intractable case of out-of-distribution detection. By establishing a theoretical framework from the perspective of feature learning, the authors try to figure out when the out-of-distribution data can be detected by post-hoc methods. The authors try to validate their findings by conducting experiments on synthetic datasets and a subset of ImageNet.

**Strengths:**

- Previous works have shown that the current definition of out-of-distribution is ambiguous. Thus the topic of this paper is important and urgent for out-of-distribution detection/generalization community.

**Weaknesses:**

- Impractical assumptions. The authors assume that the features of ID and OOD data are linearly seperated in low-dimensional spaces in a Gaussian like manner. Thought the authors highlight a few times that simplification does not compromise the validity of the conclusions, it's difficult for the reviewer to build confidence in the theoretical results based on such strong assumptions. Still the Gaussion assumption can be tolerated and widely-used in theoretical analysis, the linear separatability is rather strong, rarely-used and violates several existing works [1] according to my best knowledge.
- Non-surprising results. The main results of this paper is not surprising for me. In Corollary 1, the authors claim that if the feature vector of an OOD distribution is similar to ID distribution, it becomes intractable for post-hoc OOD detection method. The results is rather intuitive and straightforward. I am not sure if such a theorem is trivial and just formally presents some results that has been widely-known in this research field.
- Indequent experiments. The experimental results is not persuasive enough. The authors only conduct experiments on synthesized datasets and a subset of ImageNet-1K instead of the whole ImageNet OOD detection benchmark.
- Do not inspire new method. Continue to the aforementioned points, the theoretical results are derived under a rather strict assumption yet do not inspire novel OOD detection methods. Thus I generally feel the overall contribution of this paper does not meet the expectation of ICLR community.


[1] Distributions of Angles in Random Packing on Spheres, JMLR'13

**Questions:**

My main concerns lies in the strict assumption used in this manuscript and the contribution of the theoretical results.
-  Could the author provide some related works that involve the same assumptions in OOD detection/generalization or representation learning research field?
- Could the author mention a few applications of the established theory? What insight does it convey to inspire novel OOD detection method, or further research directions?

---

### Official Review · Reviewer_PMsY · 2024-11-04

**Soundness:** 2
**Presentation:** 3
**Contribution:** 2
**Rating:** 5
**Confidence:** 3

**Summary:**

This paper addresses ambiguities in current definitions for OOD detection by proposing a more precise separation of "Semantic Space" and "Covariate Space" within the ID dataset. The authors introduce a "Tractable OOD" setting that ensures distinguishability between OOD and ID distributions, focusing specifically on post-hoc detection methods that leave model training unchanged. Through theoretical analysis and experimental validation, including synthetic and ImageNet-1K data, the study demonstrates that OOD detection methods perform effectively when shifts occur within the Semantic Space but fail when shifts are limited to the Covariate Space. The contributions include formal definitions for the Semantic and Covariate Spaces, theoretical proofs on the tractability of OOD detection tasks, and an empirical evaluation showing the significance of these distinctions in improving OOD detection reliability.

**Strengths:**

1. The paper provides a precise redefinition of Semantic and Covariate Spaces, which clarifies ambiguities in existing OOD detection protocols.
2. Theoretical analysis is rigorously conducted, offering proofs on the intractability of OOD detection under certain shifts, which enhances the understanding of post-hoc OOD methods' limitations.
3. Extensive experiments on synthetic and high-dimensional data (ImageNet-1K) demonstrate the practical necessity of the proposed definitions and validate the theoretical findings, showcasing the real-world impact of the authors' approach.

**Weaknesses:**

1. The assumptions of linear separability and Gaussian distributions may not hold in many practical OOD detection applications, affecting the model's robustness​.
2. Experimental validation primarily focuses on one real dataset, which may not fully capture the complexity of real-world OOD detection tasks.
3. Experiments on OOD detection are evaluated only on AUROC, results on AUPR and some other metrics are missing.
4. The model's sensitivity to covariate versus semantic shifts is limited, which constrains its effectiveness in diverse OOD environments​.
5. The paper assumes data samples are decomposed into two spaces. Which part of the paper ensures the encoded semantic and covariate representations can be accurately divided into semantics and covariates?

**Questions:**

see weaknesses

---

### Official Review · Reviewer_6Y58 · 2024-11-04

**Soundness:** 3
**Presentation:** 3
**Contribution:** 2
**Rating:** 5
**Confidence:** 4

**Summary:**

This work proposes a more precise definition of semantic space and covariate space for ID distribution, and introduces the tractable out-of-distribution (OOD) setting that ensures distinguishability of OOD and ID distribution for post-hoc OOD detection methods. This work focuses on addressing the ambiguities in OOD detection. The proposed framework facilitates a clearer understanding of which types of shifts are tractable in OOD detection. Empirical validation demonstrates the effectiveness of the proposed definitions and framework.

**Strengths:**

1. This work focuses on addressing the ambiguities in the definitions of semantic shift and covariate shift in the filed of OOD detection, tackling an important and challenging problem.

2. The paper provides clear definitions and theoretical analysis that justify the proposed definition and concepts.

3. The writing is clear and well structured.

3. Theoretical analysis and empirical validation demonstrate the effectiveness of the proposed framework.

**Weaknesses:**

1. The concept of tractable OOD has been discussed in prior work[1], and the idea of semantic shifts and covariate shifts has been discussed in work [2], which limits the originality of this paper's contribution. A more detailed discussion and comparison with these literatures would strengthen this work.

2. The empirical evaluation of this work mainly focuses on the synthetic data and ImageNet-1K dog subsets as semantic data. Extending the evaluation of the proposed framework on other real-world practical datasets would enhance the applicability of the this paper.

3. The theoretical analysis of this work primarily relies on assumptions of Gaussian distribution of the ID data, and the linear separability of representative feature vectors in the input space. These assumptions may limit the application to more complex real-world environments.

Reference:

[1] Is Out-of-Distribution Detection Learnable? NeurIPS 2022.

[2] Feed Two Birds with One Scone: Exploiting Wild Data for Both Out-of-Distribution Generalization and Detection. ICML 2023.

**Questions:**

Refer to detailed suggestions in Weaknesses section.

---

### Official Review · Reviewer_2VQy · 2024-11-05

**Soundness:** 2
**Presentation:** 2
**Contribution:** 2
**Rating:** 3
**Confidence:** 4

**Summary:**

This work proposed a refined framework for OOD detection that introduces clear definitions for Semantic Space and Covariate Space. By establishing these definitions, the paper addresses a critical limitation in existing OOD settings, where certain distributions are indistinguishable from in-distribution samples due to ambiguous shifts. It further introduces the concept of "Tractable OOD," ensuring OOD detection is feasible for post-hoc methods, accompanied by theoretical analysis and experimental validation.

**Strengths:**

1. The manuscript provides a rigorous theoretical foundation for OOD detection by defining Semantic and Covariate Spaces, clarifying when OOD tasks become tractable or intractable.

2. The proposed "Tractable OOD" setting addresses a fundamental flaw in existing OOD frameworks, making it a valuable advancement for post-hoc OOD detection methods.

3. The manuscript demonstrates its theoretical findings through comprehensive experiments on both synthetic and real-world datasets, reinforcing the practical applicability and validity of the proposed definitions.

**Weaknesses:**

1. The manuscript’s reliance on simplifying assumptions, such as linear separability and Gaussian distributions, may limit its applicability to complex, high-dimensional data in real-world Big Data settings. Additionally, it lacks a thorough discussion on the scalability and implementation challenges of the proposed "Tractable OOD" setting in deep learning models.
 2. The experiments consider only one dataset, which is insufficient for drawing robust conclusions.
 3. The code is not available; it would be beneficial for the authors to provide a link to their code repository.

Detailed Comments:

 This paper identifies a key drawback in current OOD detection theory and introduces an intriguing framework to decompose the OOD detection problem for theoretical analysis. However, despite its innovative approach, the present version lacks practical impact on real-world applications. The quality of this paper could be greatly enhanced by incorporating the following improvements:
 1. The manuscript’s theoretical analysis is built on simplifying assumptions, such as linear separability and Gaussian-like distributions, which, although useful for analysis, may limit its generalizability to real-world scenarios involving high-dimensional, non-linear data. This reliance on basic assumptions reduces the applicability of the findings to complex settings typical in Big Data applications, where more flexible models like mixture distributions or non-parametric methods often better capture the structure of ID distributions. Furthermore, the paper does not address the practical challenges or scalability concerns of implementing the proposed "Tractable OOD" setting in deep learning models for high-dimensional data, where Gaussian assumptions might be less informative. Expanding on these limitations and exploring potential extensions or architectural modifications to accommodate more complex data structures would enhance the relevance and practicality of the approach, particularly for diverse Big Data environments.

2. Although this paper primarily focuses on a new definition of semantic shift and its theoretical foundations, using only one real-world dataset limits the strength of its conclusions, given the diverse forms of covariate shift, such as domain shift (change in the data domain or context), feature distributional shift (distribution of individual features changes between training and testing), temporal shift (changes in data distributions over time), etc. To enhance the robustness of the findings, the authors might consider including additional real-world datasets in the experiments. For instance, the "Waterbirds" dataset [1] could be valuable for testing spurious correlations—a special case of feature distribution shift where irrelevant features influence model predictions.

3. The manuscript lacks explicit information on the availability of code or dataset access links, which are essential for reproducibility and community engagement. Providing a link to a code repository or outlining plans to make the code available upon publication would be beneficial.

[1] Sagawa, Shiori, et al. "Distributionally robust neural networks for group shifts: On the importance of regularization for worst-case generalization." arXiv preprint arXiv:1911.08731 (2019).

**Questions:**

Q1: How could your proposed definitions and theorems support deploying state-of-the-art OOD detection methods in real-world scenarios? Please clarify the practical relevance of your theoretical framework and discuss how it could enhance or integrate with existing SOTA OOD techniques to address real-world challenges.

Q2: The experiments consider only one real-world dataset, "IMAGENET DOGS." The authors' results will be more convincing if they consider more real-world datasets, such as the Waterbirds [1] or other benchmark datasets appropriate for this context.

Please see my comments on the weaknesses as well.

---

### Comment · Area_Chair_q8BT · 2024-11-22

Dear Authors and Reviewers,

The discussion phase has passed 10 days. If you want to discuss this with each other, please post your thoughts by adding official comments.

Thanks for your efforts and contributions to ICLR 2025.

Best regards,

Your Area Chair

---

### Note · Authors · 2024-11-25

**Comment:**

We sincerely appreciate the valuable feedback provided by the reviewers, which helps us identify some shortcomings in our work. Given that the manuscript requires significant improvements, we decide to withdraw it in order to further refine the paper. In the future, we will reconsider submission after making substantial revisions.

**Withdrawal Confirmation:**

I have read and agree with the venue's withdrawal policy on behalf of myself and my co-authors.